# The Ferritin, Hepcidin and Cytokines Link in the Diagnoses of Iron Deficiency Anaemia during Pregnancy: A Review

**DOI:** 10.3390/ijms241713323

**Published:** 2023-08-28

**Authors:** Yvonne Chibanda, Matthew Brookes, David Churchill, Hafid Al-Hassi

**Affiliations:** 1Research Institute in Healthcare Science, University of Wolverhampton, Wolverhampton WV1 1LY, UK; 2Gastroenterology, The Royal Wolverhampton NHS Trust, Wolverhampton WV10 0QP, UK; 3Obstetrics, The Royal Wolverhampton NHS Trust, Wolverhampton WV10 0QP, UK

**Keywords:** serum ferritin, hepcidin, inflammation, iron replacement therapy, pregnancy

## Abstract

Following a diagnosis of iron deficiency anaemia in pregnancy, iron supplements are prescribed using UK guidelines; however, despite this, the condition remains highly prevalent, affecting up to 30% of pregnant women in the UK. According to the World Health Organisation, it globally accounts for 45% in the most vulnerable groups of pregnant women and infants (<5 years old). Recently, the efficacy of iron replacement therapy and the effectiveness of current standard testing of iron parameters have been reviewed in order to evaluate whether a more accurate diagnosis can be made using alternative and/or supplementary markers. Furthermore, many questions remain about the mechanisms involved in iron metabolism during pregnancy. The most recent studies have shed more light on serum hepcidin and raised questions on the significance of pregnancy related inflammatory markers including cytokines in iron deficiency anaemia. However, research into this is still scarce, and this review aims to contribute to further understanding and elucidating these areas.

## 1. Introduction

Iron, from the Anglo-Saxon word *iren*, is the most abundant element on earth. Biologically, iron is multifunctional with central roles in DNA synthesis, oxygen transport, the electron transport chain function and red cell production [1]. However, free iron can cause protein, lipid and nucleic acid damage via the generation of reactive oxygen species (ROS) such as the hydroperoxyl radical (HOO·), hydroxyl radicals (^−^HO·) and superoxide anions (HO^−^) [2]. Hence, tight regulation of the ‘closed’ iron circulation is required to ensure the prevention of iron overload and toxicity, as there is no mechanism of iron excretion other than through sweat and natural cell shedding (every 3–4 days) excreted in faeces [2].

Iron metabolism: Biological iron exists in two states, haemiron (ferrous Fe^2+)^ and non-haem (ferric Fe^3+^), with soluble and insoluble functionality, respectively. Once ingested, ferric iron travels to the duodenum and proximal jejunum of the small intestine where absorption in the circulation takes place from the apical to the basolateral end of the enterocytes. Ferric iron is converted into ferrous iron by duodenal cytochrome b (cytb) and Vitamin C ferriductase in a reduction reaction [3]; this then allows for its intake into the enterocyte via cellular membrane bound divalent metal transporter 1 (DMT1) [4]. Intracellular ferrous iron will either be oxidised back into ferric iron for storage as ferritin or released via ferroportin (FPN) at the basolateral cell surface, depending on systemic requirements. This is signalled by hepcidin, which regulates circulating iron by controlling FPN function, although intracellular iron levels are modulated by IRP1 and IRP2.

At the point of FPN mediated release, the enzymes ceruloplasmin and hephaestin catalyse the oxidation of ferrous iron back to ferric iron, which, at a pH of 7.4, has a high affinity to associate with circulating apotransferrin. This results in the formation of a diferric transferrin complex (Tf) which carries a maximum of two ferric iron molecules. Diferric transferrin binds to transferrin receptors (TfR1 or TfR2) found on the cell surface of most cells at varying levels including macrophages in the spleen, hepatic cells and, more specifically for this project, TfR1, which is highly expressed in placental tissue and erythroid cells, which make up the majority (75%) of iron consumption. To our knowledge, there is no evidence on the expression of TfR2 in the placental tissues. The reason for this could be because the TfR2α, the extracellular domain of TfR2, is highly homologous to TfR1 although with lower Tf binding affinity compared with TfR1 [5]. Hence, future investigation into the expression of TfR2 in placental tissue is needed. 

The diferric Tf and TfR bind via a carbonate, forming a Tf–TfR complex; this is engulfed via receptor-mediated endocytosis [6]. The endosomes’ hydrogen ion pumps (H+ ATPases) cause an intravesicle drop in pH to 5.5, which allows dissociation of the ferric iron from the differic Tf. Ferric iron will be further reduced into ferrous iron by STEAP as explained below. The free ferric iron is then used by cells for oxygen transport, energy metabolism and DNA synthesis.

The now iron-free Tf, apotransferrin, travels back to the cell surface to be released back into circulation to ‘pick up’ more available ferric iron.

Haemiron (from myoglobin or haemoglobin) is also taken up by the enterocytes but via the Haem Carrier protein 1 (HCP1) at the apical enterocyte. Once in the cell, Haem-Fe^2+^ is broken down via haem oxygenase (HO-1), releasing the Fe^2+^, which is either converted to Fe^3+^ for storage (within ferritin) or release via FPN in the same way as described for Fe^3+^.

The Haem component of HaemFe^2+^ is thought to be broken down into bilirubin or released via breast-cancer-resistant protein (BCRP) and/or feline leukaemia virus subgroup C (FLVCR) at the basolateral of the enterocyte.

### 1.1. Iron and Pregnancy

#### Iron Transport: Maternal to Placenta to Foetus

The human placenta is a haemochorial organ comprising several cell types including syncytiotrophoblast, endothelium, mesenchymal, fibroblast and Hofbauer cells.

The syncytiotrophoblast cells, located between maternal and foetal circulation, are responsible for nutrient transfer and are where placental iron transport occurs. Since placental and foetal iron is obtained from the maternal circulation, it is imperative that maternal iron supply and storage is adequate to accommodate pre- and post-natal demand.

The high concentration of transferrin receptors on the apical membrane of the syncytiotrophoblast creates a hierarchy of iron distribution during pregnancy: the placenta, foetus and mother are prioritised in that order.

The human placenta has a single layer of syncytiotrophoblast; the apical membrane (maternal side) possesses numerous TFR1 and the basolateral (foetal side) possesses FPN. In the same internalisation/engulfing process described in the aforementioned ‘General iron metabolism’ section, the Fe^3+^ dissociated from the transferrin–TFR1 receptor complex within the endosome, it is then thought to be reduced by ferriductases STEAP3 and/or STEAP4 [7]. The Fe^2+^ is then thought to be transported out of the endosome via ZIP8, ZIP4 or DMT1. This then allows the endosome to return to the apical membrane, exposing and releasing apotransferrin at pH 7.4 back into maternal circulation.

The freed Fe^2+^ is either (i) stored as ferritin within the syncytiotrophoblast or (ii) chaperoned by PCBP1 and 2 through to the basolateral FPN, where it is thought to be oxidised by ferroxidase ZP, CP or HEPH; this is then released to foetal Tf [8,9,10].

### 1.2. Iron Deficiency Anaemia

Iron deficiency is a condition that results from long-term depletion of iron stores. It has several causes including haemorrhage, inadequate dietary intake and malabsorption syndromes. Adequate levels of maternal iron are vital throughout pregnancy to also account for increased demand from the growing foetoplacental unit [11]. The effects of iron deficiency on its functions depend on its severity. Severe iron deficiency affects healthy erythropoietin-mediated erythropoiesis and causes anaemia: a reduction in haemoglobin concentration and red blood cell count [12]. Therefore, iron deficiency can present with or without anaemia. Although anaemia can be caused by other deficiencies such as B12 and folic acid, iron deficiency is by far the most common cause during pregnancy [13].

Despite numerous options of iron replacement therapy [14], IDA remains highly prevalent across the globe and, according to the World Health Organisation (WHO), 2019, accounts for 45% in the most vulnerable groups of pregnant women and infants (<5 years old). The importance of this is stipulated in the WHO’s 2025 target to halve global anaemia in women of reproductive age. Iron deficiency without anaemia affects 18% of pregnant women with 7% in the first trimester and 30% in the third trimester [15].

IDA during pregnancy has been linked to aberrant neonatal neurodevelopment, e.g., autism spectrum disorder and attention deficit hyperactivity disorder [16,17,18,19]. Excess iron replacement therapy is thought to be associated with preterm birth and infection [19]. Essentially, there are negative pregnancy outcomes associated with haemoglobin levels lower 110 g/L or greater than 130 g/L. Maintaining adequate levels of iron throughout pregnancy is challenging due to changes in iron demand at each trimester; more dietary iron is absorbed in the third trimester (7.5 mg) than in the first trimester (0.8 mg) [20].

### 1.3. Diagnosis

Accurate diagnosis, treatment and maintenance of iron and haemoglobin levels in pregnant women are a work in progress. This is because current diagnostic tests are not nationally standardised [21], and interpretation is problematic in the presence of inflammation and infection [22]. Treatment is dependent on diagnosis; therefore, reviewing the limitations to interpretation of standard tests is necessary to assess their effectiveness. Current iron testing includes serum ferritin, which is an acute phase reactant whose levels are affected by inflammation; Tf saturation, which fluctuates due to diurnal variation of serum iron; and serum iron, which decreases with malignancy, infection and inflammation and increases with liver disease [23]. Haemoglobin measurement tends to be more robust however, and establishing the cause of anaemia is important, i.e., caused by inflammation or iron deficiency [19]. This would help prevent unnecessary prescription of iron replacement therapy.

Serum ferritin: Serum ferritin (diagnostic cut off in pregnancy ≤ 30 ng/mL) has been the most commonly used non-invasive test in diagnosing IDA when interpreted with haemoglobin levels. However, this is complicated in the presence of inflammation and lacks evidence in pregnancy. During an inflammatory response to infection, cytokines such as interleukin 1 beta prevent the release of cellular iron into the circulation. This occurs by upregulating the expression of heavy chain ferritin (in liver cells), resulting in increased iron storage. This deprives the invading pathogen of the hosts’ iron which it needs to thrive [24]. It has been suggested that the increase in maternal cytokines during the acute inflammatory response is more detrimental to foetal neurodevelopment than (some/specific) the invading pathogen itself [25]. In addition, pathogens such as Salmonella are capable of hepcidin-independent FPN downregulation within macrophages [26]. This further complicates diagnosis because it results in iron delocalisation (i.e., accumulation in secretions and tissue) while deficient in blood, akin to hereditary haemochromatosis phenotype. 

Serum hepcidin is a 21st century discovery: Hepcidin is a hepatic-produced 25-amino acid peptide protein that functions as the main systemic mediator of iron homeostasis. It was first discovered in human urine and serum and described in the year 2000. The functional hormone, hepcidin-25, is the cleaved product of prohormone (60 amino acids), which is a divided product of preprohormone (84 amino acids) encoded by the HAMP gene. Within circulation, hepcidin-25 is bound to α-2-macroglobulin and albumin. Functionally, hepcidin binds to, internalises and degrades FPN, the cell membrane bound gatekeeper of transcellular iron transport. FPN is found on enterocytes, hepatocytes, reticuloendothelial macrophages and adipocytes. Therefore, an increase and decrease in levels of hepcidin causes opposing iron availability. Hepcidin is upregulated and downregulated by several factors illustrated in Table 1; a complete lack of hepcidin causes juvenile haemochromatosis, and overexpression causes iron-restricted anaemia.

Studies on hepcidin to see if it is a superior test over the current standard iron tests have not been conclusive. Research results vary in showing it to be comparable [27], superior [28] or advantageous only when it is used in conjunction with another test, such as haemoglobin concentrations [29].

Initially linked to iron homeostasis in 2001, there is yet to be a standardised, robust diagnostic laboratory test for hepcidin. Current options for the measurement of hepcidin consist of testing levels in urine or serum, both of which show correlating levels (excluding renal failure). Techniques such as enzyme-linked immunosorbent assay (ELISA) and mass spectrometry have been employed in research to measure hepcidin, with the first commercial ELISA test having been trialled in 2011 by Geerts, Vermeersch and Joosten [30], although on a geriatric cohort of patients not in pregnancy. There are two main differences between the two techniques: cost and specificity. Mass spectrometry has high specificity in distinguishing the different hepcidin isoforms; however, it is not cost-effective; ELISA and competitive enzyme-linked immunosorbent assay (C-ELISA) are cost-effective and relatively efficient in measuring hepcidin levels but do not distinguish between isoforms, rather portraying total hepcidin. Affordability and turnaround times are of importance when integrating such a test within the healthcare setting.

Ratios: In terms of IDA diagnosis, correlation of serum ferritin and hepcidin [11] does not necessary make serum hepcidin superior to serum ferritin, and it might not be cost-effective to use both tests. However, its usefulness may be when used in conjunction with other tests or as ratios to predict risk of IDA. Likelihood ratios, indices or quotients (referred to hereafter as ratio) are calculated using standard test results that have deter-mined sensitivity and specificity in ratio format to predict the likelihood of a positive or negative diagnosis in pathology. The idea of using ratios is not a new one [31] but would be a novel approach in predicting IDA risk in pregnant women. Most iron-status-related ratios have been studied in cohorts which are different to pregnancy, such as cirrhosis [32], blood donors [31], haemoglobinopathies [33], neurodegeneration [34] and athletes [35]. However, applying this approach to pregnancy would potentially be beneficial to further guide treatment allowing for a more cost-effective, non-invasive and preventative strategy. 

Considering that both serum ferritin and hepcidin levels are affected by inflammation [36,37], it could therefore be useful to include cytokines and/or soluble TfR testing to aid differential diagnosis. One such example is an animal study by Sangkhae et al., 2020, which included a ratio where PIDI determines the risk of iron deficiency to the foetus by measuring ferroportin and TfR1 levels on the placenta following parturition [38]. However, this has not tested on human samples, and more research is needed in this area.

Another example on using ratios in the diagnosis of iron deficiency is the Mentzer index, used to distinguish the likelihood of iron deficiency versus a haemoglobinopathy. This is calculated from dividing the mean corpuscular volume (MCV) by the red cell (RBC) count from a full blood count result. If the value is above 13, the likelihood is ID, whereas lower than 13 is indicative of beta thalassemia [39]. 

Total iron-binding capacity (TIBC) is also a diagnostic tool for IDA and iron overload. TIBC values are also derived from the calculation of the ratio of serum iron and total iron-binding capacity, which is an indirect measure of transferrin levels [40].

Under similar principles, measuring the ratios of iron metabolism proteins can be applied in the early diagnosis of incipient IDA in pregnancy. It may be useful to investigate ratio combinations using different iron metabolism markers, such as hepcidin-to-ferritin ratio [41], hepcidin-to-Tf ratio and ferritin-to-Tf ratio [42]. 

### 1.4. Inflammation and Cytokines

Although iron deficiency is the primary dietary cause of anaemia, it is important to determine this by ruling out other causes such as inflammation or malignancy. 

There is a close interconnection between iron metabolism and immunity. This is largely due to the production of cytokines from immune cells including T cells and macrophages under certain physiological conditions [43]. These cytokines regulate the iron-handling proteins involved in the uptake, storage and release of iron [44]. For instance, interleukin-4 (IL-4), IL-10 and IL-13 increase the cellular uptake of iron via upregulation of TfR mRNA expression on THP-1 monocytes, and IFN-γ reduces iron uptake by decreasing TfR mRNA expression on the same cells [45,46]. In addition, IL-1β, IL-6 and TNF-α induce ferritin synthesis [47], and IL-22 increases hepcidin expression independent of IL-6 [48]. 

The role of pregnancy-related type 1 T helper cells (TH1, pro-inflammatory) and type 2 (TH2, anti-inflammatory) cytokines during each trimester needs further investigation to better understand their role in iron metabolism (summarised in Figure 1). However, increasing evidence suggest that, in the first trimester, there is a significant upregulation of TH1 cytokine production including IL-1, IL-6, IL-8 and tumour necrosing factor alpha (TNF-α), whereas the second trimester is characterised by the prevalence TH2 cytokines including IL-10 and IL-4 and transforming growth factor beta (TGF-β). In the third trimester, a balance between TH1 and TH2 cytokine production is needed to protect against infection and maintain the pregnancy [49]. Hence, moderate inflammation during pregnancy is physiological [50,51]. However, some cytokines such as INF-γ, IL-2, IL-6 and TGF-β are pleiotropic, having dual functionality [52,53]. 

Inflammation is a complex process implicating the interaction of many inflammatory mediators. These include recruitment and activation of various inflammatory immune cells such as neutrophils, basophils, T cells, macrophages and dendritic cells as well as production of pro-inflammatory cytokines and mediators of inflammation such as IL-6, IL-8, IL-12, IL-23, TNF-α, IFN-γ and hepcidin, and anti-inflammatory cytokines including IL-4, IL-10 and TGF-β. Interactions between all or some of these mediators can cause inflammation and tissue damage [54]. Therefore, establishing a test that includes the ratio combination between pro- and anti-inflammatory cytokines concurrently with a ferritin test may potentially also be useful as a tool to investigate IDA in pregnancy and to distinguish whether the cause is due to inflammatory conditions. Other inflammation markers such as C-reactive protein and α_1_-acid glycoprotein can also be added to the combined test as described in the BRIDA project [55].

During early pregnancy, approximately 20–30% of leukocytes infiltrating the decidua are macrophages [56]. Based on their function and cytokine production profile, macrophages are classified as M1 or M2 phenotype [57]. The M1 phenotype is pro-inflammatory and produces pro-inflammatory cytokines such as IL-6, IL-12 and TNF-α, whereas M2 is anti-inflammatory and produces anti-inflammatory cytokines such as IL-4, IL-10 and TGF-β [58]. In healthy pregnancy, the majority of macrophages on the maternal–foetal interface are of the M2 phenotype. M2 macrophages are involved in tissue remodelling and induce tolerogenic immune responses and an immune-suppressive environment during the pregnancy period [59]. However, a recent study has shown that both phenotypes exist on the maternal–foetal interface, presumably to work in concert to protect this interface from infection but also to promote foetus growth and induce angiogenesis [59]. It is likely that the profile of cytokine production by the macrophage subsets is not fixed but regulated by exposure to pregnancy microenvironment that can alter cytokine production and the generated appropriate immune responses. 

However, under pathological conditions, for example in pregnant women with pre-eclampsia, during the second trimester, the transition to the M2 profile is blocked, and the M1 pro-inflammatory immune profile is activated, producing elevated levels of the pro-inflammatory cytokines IFN-γ, TNFa and IL-6 and reduced levels of IL-4 and IL-10 [60]. Interestingly, a study by Brunacci F et al., 2018, has shown that total serum iron was significantly higher in third-trimester pregnant women with pre-eclampsia compared to pregnant healthy control women [61].

During pregnancy, the occurrence of anaemia may not be due solely to absolute iron deficiency; it could occur as a result of a reduction in iron functional immobilisation and its bioavailability for cellular metabolism. This is known as anaemia of chronic disorders or anaemia of chronic inflammation (ACI) [49]. 

Prolonged upregulation of pro-inflammatory cytokines including IL-1, IL-6 and interferons, frequently prevalent in women with chronic diseases such as type 2 diabetes, also causes dysregulation of iron metabolism due to an increase in hepcidin production by the liver and a decrease in ferroportin as well as reduction in erythropoietin function, which leads to the induction of ACI [62,63,64]. Thus, the increase in pro-inflammatory cytokines as a result of chronic disease leads to iron ‘delocalisation’ or iron overload in the tissues and macrophages and iron deficiency in the circulation rather than absolute iron deficiency as explained by Rosa et al., 2017 [65].

Abnormally high inflammation can be acute, i.e., sudden response to infection or chronic, which is long-term. Aberrant cytokine levels have been linked to adverse foetal neurodevelopment outcomes lasting into adulthood. For example, IL-6 is associated with pregnancy implantation and embryogenesis; raised IL-6 levels have been linked to behaviour abnormalities in offspring of animal models [66,67,68,69]. Research into treatment of ID and IDA using bovine milk derivative lactoferrin to target lowering IL-6 is promising, as it has been shown to normalise haemoglobin levels in IDA pregnant women with thalassemia and other pathologies [64]. 

Although research into the role of these cytokines is scarce, there is evidence to suggest a central role in adverse pregnancy outcomes when in the presence of surplus iron [11,70]. For example, animal studies have illustrated that iron deficiency has a lowering effect on the expression of TNF-α, needed for neonatal growth and development [71]. It is well known that pro-inflammatory cytokines increase the synthesis of both ferritin and hepcidin [72]. Therefore, the gradual decline in ferritin and hepcidin levels during the first trimester where the pro-inflammatory cytokine levels are high could be inducing a positive feedback mechanism for iron homeostasis.

**Figure 1 ijms-24-13323-f001:**
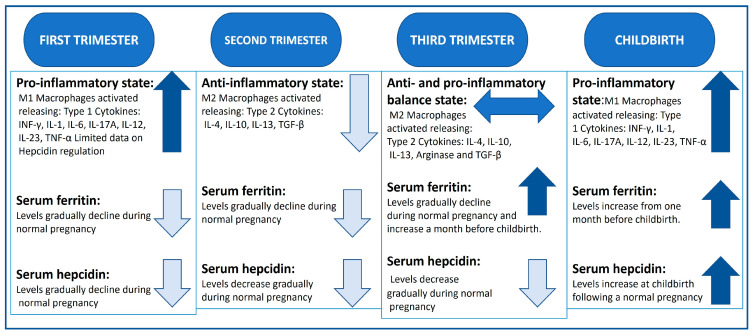
Summary of the inflammatory profile, serum ferritin and hepcidin levels in normal pregnancy [46,70,73]. Levels portrayed as ↑ increased, ↓ declining levels, ↔ balance of both.

### 1.5. Treatment

Failing dietary iron supplementation in the reversal of iron depletion induced anaemia; pharmaceutical intervention is prescribed with oral iron intake being the first line of treatment; serum ferritin checked for women with known haemoglobinopathies prior to prescribing treatment [74]. The WHO recommended daily dose is 30–60 mg elemental iron. A higher dose may be administered to non-responders who do not experience side effects to the first dose. Lowered intermittent doses are administered as a secondary step to combat gastric irritation/nausea commonly faced because the more iron is available in the tablet, the greater the risk of side effects including gastric irritation [75].

An increase in reticulocyte count is expected 3–5 days post-ingestion followed by an increase in haemoglobin concentration (1–3 weeks) (British Society of Gastroenterology guidelines). 

### 1.6. Oral Iron Treatments

Commonly used oral iron treatments include preparations of ferrous (Fe^2+^) salts, such as ferrous sulphate, gluconate and fumarate, and ferric (Fe^3+^) salts, such as ferric polymaltose, sulphate, citrate, etc., some of which are available in tablet and/or syrup form. Ferrous salts have superior absorption in comparison to ferric salts as they are easily transported via enterocyte apical membrane. At low gastric pH, ferrous irons dissociate from ferrous fumarate, allowing for absorption by the enterocytes via general metabolism route. Ferrous bisglycinate is thought to have increased compliance with little to no side effects due to chelation with the amino acid glycine [76]. 

### 1.7. ACI Treatment

The fact that ACI is characterised by iron dislocation and iron overload in tissues and macrophages makes traditional oral and intravenous iron treatment, including ferric sulphate and ferric carboxymaltose, a challenging issue. This is because the iron supplementation will increase IL-6 and decrease ferroprotein production, which, in turn, may exacerbate iron overload and will potentially increase the risk of infection [77]. 

Lactoferrin is a glycoprotein that is produced in human secretions, including saliva, and mucosal areas [78]. Lactoferrin is an anti-inflammatory agent that has a protective function against mucosal infection and microbial colonisation [78,79]. Hence, the bovine milk derivative lactoferrin has been utilised as a natural therapy for the treatment of IDA and ACI [64]. 

The mechanism of action of bovine lactoferrin treatment is via reducing IL-6 and hepcidin production levels and upregulating the production levels of ferroportin [63]. addition, Cutone A et al. have shown that increasing levels of IL-6 have downregulated the expression levels of ferroportin in macrophages and that the treatment with bovine lactoferrin reversed this effect and upregulated ferroportin [80].

In pregnant anaemic women with various pathological conditions, bovine lactoferrin treatment was significantly more effective than ferrous sulphate in treating ACI. For instance, in pregnant women with minor β-thalassemia, a recessive genetic disorder that reduces haemoglobin production, bovine lactoferrin has significantly increased total serum iron and reduced the production of IL-6 compared to ferric sulphate [63]. 

Taken together, this evidence has shown that that bovine lactoferrin is effective in ACI treatment during pregnancy.

### 1.8. Intravenous (IV) Iron Therapy

IV therapy may be recommended in those non-responsive to any oral treatment. Commonly used IV iron treatments include iron dextran, ferric carboxymaltose (Ferinject), iron sucrose (Venofa), sodium ferric gluconate, ferumoxytol and iron isomaltoside (Monofer). Although it has been reported that iron dextran infusion resulted in anaphylaxis, anaphylactoid reaction and upper airway angioedema, the current iron formulations prove to be efficacious and safe with approximately 3.6 in 1000 women having serious adverse effects [81]. It has also been reported that IV therapy is more effective in replenishing iron stores compared with oral iron treatment [82]. 

### 1.9. Iron Replacement Alternatives

Blood transfusions are reserved for more severe urgent symptomatic cases. Treatment with erythropoietin, adjuvant recombinant human erythropoietin (rHuEPO) has been documented as having good compliance in combatting anaemia, although further studies are needed to validate its efficacy and safety [83].

Considerations for iron absorption include gastric acid stability, types of iron given (slowed or delayed release formulas may bypass optimal absorption at enterocytes), patients’ iron stores (increased absorption if deficient), dose administered, elemental iron (free iron in preparation), bioavailability (absorbable into functional circulation) and dietary inhibitors such as phytates, polyphenols, calcium, animal proteins, soy protein and oxalic acid. Amino acids and ascorbic acid enhance iron absorption; however, this is still reduced in the presence of phytates [84].

### 1.10. Hepcidin Agonists and Antagonists

Treatment using hepcidin agonists and antagonists may prove useful dependent on a harmonised hepcidin test. 

The normal levels of iron concentration in the plasma are within 10–30 μM, and long-term changes in these levels can cause diseases associated with iron disorders.

Increasing levels of hepcidin in the serum leads to a lower iron levels and increased pro-inflammatory cytokines production from inflammatory immune cells, which subsequently leads to anaemia of inflammation. Other diseases such as chronic kidney disease (CKD), inflammatory bowel disease, iron refractory and cancer are all associated with high hepcidin levels [85,86]. Currently, a novel group of hepcidin antagonist drugs is in late-stage clinical control trials and could be beneficial in lowering hepcidin production. For example, LY3113593, a humanised anti-BMP6 monoclonal antibody, whose mechanism of action is to neutralise BMP6, was shown to be effective in the treatment of anaemia in patients with CKD. Furthermore, the monoclonal anti-IL-6 receptor antibody has been shown to be effective in the treatment of anaemia by significantly reducing serum hepcidin in rheumatoid arthritis patients [85]. However, thus far, there is no evidence on the efficacy and safety of these drugs in the treatment of hepcidin-elevation-related disorders in pregnant women.

Hepcidin deficiency leads to unregulated ferroportin activity, thus cellular transport of iron into the plasma increases, which can cause chronic liver disease due to cellular iron overload, IDA, congenital dyserythropoietic anaemias, sideroblastic anaemias, β-thalassaemia and myelodysplastic syndromes [86,87]. Hepcidin agonists including BMP6; VIT-2763, which is a ferroportin blocker; antisense oligonucleotides; and siRNAs that inactivate matriptase-2 to increase endogenous hepcidin production are promising therapeutic agents that reduce iron overload [88]. However, these hepcidin agonists are not yet available to patients or still in the control trials stage. In addition, there are no data yet on their usage and effectiveness in pregnancy.

## 2. Conclusions

From ancient-history discoveries of biological intracellular iron to recent history of iron status assessment using the invasive technique of bone marrow investigation, which has for the most part been replaced by serum ferritin amongst other tests, current methodologies still create vacancies for future improvement and/or implementation of newer protein testing, for example measuring the ratios of pro- and anti-cytokines and the functional implication of the glycosylation pattern of TfR1 in pregnancy and non-pregnant individuals, which can help differentiate between IDA and ACI [89].

Antenatal care guidelines in the UK allow for prevention of incipient IDA [76]. Iron demand is greater during the third trimester [90]; congenital IDA in the newborn is difficult to treat [91] and is linked to maternal IDA in the third trimester [92]. Therefore, prevention would be better than a cure. However, in cases more severe and/or problematic to treat, tailored diagnosis and treatment would still be needed in order to maintain haemoglobin levels within a therapeutic range. However, the following questions remain:How do specific pregnancy pro-inflammatory and anti-inflammatory cytokines affect iron metabolism in iron deficiency anaemia?Because cytokines work in a dynamic system, what cytokine combination can be revealed per individual that can predict treatment responses?Which combination of tests gives reliable ratios to predict iron deficiency anaemia during pregnancy?How can we standardise serum ferritin and hepcidin reference ranges?

Answering these questions will potentially establish the basis for individualised diagnosis and treatment of IDA in pregnant women.

## Figures and Tables

**Table 1 ijms-24-13323-t001:** An illustration of factors influencing levels of circulating hepcidin.

Upregulated	Downregulated
Post partum	First, second and third trimester of pregnancy
Inflammation (Cytokine IL-6 induced) (Host protection mechanism from infection by depriving microbes of iron supply)	IDA (Remains downregulated during inflammation in severe cases)
Raised serum iron levels	Haemolytic anaemias
Oral or IV iron intake	Anaemias with ineffective erythropoiesis
Infection	Chronic liver disease
Low GFR and CKD	Chronic HCV infection

## Data Availability

Not applicable.

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
