# Peer review of "The Ferritin, Hepcidin and Cytokines Link in the Diagnoses of Iron Deficiency Anaemia during Pregnancy: A Review"

_ijms, 2023, doi:10.3390/ijms241713323_

Round 1

Reviewer 1 Report

The review by Chibanda et al. is mainly focused on iron metabolism in pregnant women. By describing systemic iron homeostasis and the key serum parameters to follow, this review provides many useful information about IDA diagnosis in pregnant women.

Minor points:

Lines 50-54: Transferrin receptor 2 has a regulatory role in systemic iron homeostasis. Is it known whether it is also expressed in placental tissue?

Line 41 Reference number 3 is used for?

Line 59: ferric iron is reduced into ferrous again by STEAP as described in line 85

Line 90: is it PCBP1 or 2 to act as a chaperon for ferroportin? Is it already demonstrated what ferroxidase is expressed at syncytiotrophoblast levels?

Lines 172-182 could be synthetized.

Author Response

We thank the reviewers and the editors for their insightful and helpful comments on our study, which we believe have improved the overall content, readability and relevance of this manuscript. Below we provide point-by-point responses to the comments made. The changes we have made to the manuscript are tracked in yellow.

Reviewer 1

Comments and Suggestions for Authors

The review by Chibanda et al. is mainly focused on iron metabolism in pregnant women. By describing systemic iron homeostasis and the key serum parameters to follow, this review provides many useful information about IDA diagnosis in pregnant women.

We thank the reviewer for commenting on the usefulness of the information that our review provides.

Minor points:

Lines 50-54: Transferrin receptor 2 has a regulatory role in systemic iron homeostasis. Is it known whether it is also expressed in placental tissue?

Author response: This is a very good question and we have edit our paragraph (Line 53) but we also added a new paragraph (Line 54). We cannot be certain whether TfR2 expressed on the placenta because the alpha segment of the receptor is similar to that of TfR1 and therefore more work needs to be carried out to understand the signalling mechanisms of Tf in the placenta.

Line 41 Reference number 3 is used for?

Author Response: We have deleted this reference.

Line 59: ferric iron is reduced into ferrous again by STEAP as described in line 85.

Author response: We have added this to the paragraph (Line 62).

Line 90: is it PCBP1 or 2 to act as a chaperon for ferroportin? Is it already demonstrated what ferroxidase is expressed at syncytiotrophoblast levels?

Author Response: It is PCBP1 mainly but PCBP2 can also acts as a chaperon for ferroportin. We have edited the paragraph (Line 95) accordingly.

Lines 172-182 could be synthetized.

Author Response: We have edited the paragraph (Lines 182-193).

Thank you again to the reviewer and the editors for considering our manuscript and for providing this constructive critique. We hope that you find our responses satisfactory.

Reviewer 2 Report

This manuscript presents a review of the link between ferritin, hepcidin and cytokines in the diagnosis of iron deficiency anaemia during pregnancy. The link between inflammation and iron status is important so this review is topical and of interest to readers. While the cytokine and inflammatory component has been discussed in detail, an in-depth evaluation of its link with iron status in this review is lacking currently.

The below comments are provided to help address some of this issue:

1. Iron deficiency rates - the paragraph starting on line 103 gives a brief outline of the problem of IDA in vulnerable groups but it would be helpful if the authors provided some additional context as to the scale of the issue with iron deficiency during pregnancy. For example, what's the prevalence of iron deficiency (without anaemia) across gestation in different settings, who's more vulnerable (i.e. increased risk in 3rd trimester, in vegan/vegetarian women etc.).

2. Inflammation- the authors have detailed the inflammatory/cytokine changes that occur during pregnancy but in the diagnosis section on ferritin particularly, there is no mention of the need to measure concurrent inflammatory markers to validate ferritin measurements. The fact that ferritin is an acute phase reactant is stated but very little around how to address that has been included - for example, there's no reference to the BRINDA work or the need to measure CRP or AGP concurrently with iron status indicators. This needs to tie in better with the paragraph starting on line 184 also.

3. In line 170, the potential "treatment using hepcidin agonists and antagonists" has been mentioned but no data or references to support a statement like this have been made. This is a pretty novel concept so further discussion or elaboration of the point is needed for the reader.

4. IV therapy - this concept is mentioned in the paragraph starting on line 324 but there is no consideration provided on the use of IV iron during pregnancy. There are limitations and potential risks to its use in pregnancy, although there are large trials ongoing exploring this further at the moment. As this review is focused on pregnancy, the discussion of these topics should be more focused on that population.

5. Table 1 - this table outlines the factors that result in up or down-regulation of hepcidin. The authors should also note that pregnancy itself results in a down-regulation of hepcidin, particularly in the 2nd and 3rd trimesters (see van Santen et al. 2013).

Minor Comments (formatting related):

1. Just to note that throughout the manuscript, inappropriate hyphenation of words has occurred. This is likely due to a formatting issue but I wanted to flag this now so that it can re rectified. 

2. The use of sub-headings and section headings appears to be inconsistent throughout the text, with new sections or sub-sections needing to be more clearly distinguished than in their current form (see lines 36, 93 for examples of current issue).

Author Response

We thank the reviewers and the editors for their insightful and helpful comments on our study, which we believe have improved the overall content, readability and relevance of this manuscript. Below we provide point-by-point responses to the comments made. The changes we have made to the manuscript are tracked in yellow.

Reviewer 2

Comments and Suggestions for Authors

This manuscript presents a review of the link between ferritin, hepcidin and cytokines in the diagnosis of iron deficiency anaemia during pregnancy. The link between inflammation and iron status is important so this review is topical and of interest to readers. While the cytokine and inflammatory component has been discussed in detail, an in-depth evaluation of its link with iron status in this review is lacking currently.

The below comments are provided to help address some of this issue:

  1. Iron deficiency rates - the paragraph starting on line 103 gives a brief outline of the problem of IDA in vulnerable groups but it would be helpful if the authors provided some additional context as to the scale of the issue with iron deficiency during pregnancy. For example, what's the prevalence of iron deficiency (without anaemia) across gestation in different settings, who's more vulnerable (i.e. increased risk in 3rd trimester, in vegan/vegetarian women etc.).

Author Response: Thank you to the reviewer for this comment as we agree that additional information on scale of IDA in pregnancy would be enriching to the review. Therefore, we have added more context on the prevalence of IDA (Line (108).

  1. Inflammation- the authors have detailed the inflammatory/cytokine changes that occur during pregnancy but in the diagnosis section on ferritin particularly, there is no mention of the need to measure concurrent inflammatory markers to validate ferritin measurements. The fact that ferritin is an acute phase reactant is stated but very little around how to address that has been included - for example, there's no reference to the BRINDA work or the need to measure CRP or AGP concurrently with iron status indicators. This needs to tie in better with the paragraph starting on line 184 also.

Author response: Thank you to the reviewer for pointing this out and we agree that linking the cytokine context in the pregnancy section with the diagnostic section has significantly improved the manuscript. Hence, we have edited this section and cited the BRINDA project (Line 243).

  1. In line 170, the potential "treatment using hepcidin agonists and antagonists" has been mentioned but no data or references to support a statement like this have been made. This is a pretty novel concept so further discussion or elaboration of the point is needed for the reader.

Author response: Again we are grateful to the reviewer for this comment as it has enriched the review. We have expanded on this section added the relevant references (Line 369).

  1. IV therapy - this concept is mentioned in the paragraph starting on line 324 but there is no consideration provided on the use of IV iron during pregnancy. There are limitations and potential risks to its use in pregnancy, although there are large trials ongoing exploring this further at the moment. As this review is focused on pregnancy, the discussion of these topics should be more focused on that population.

Author response: We agree with the reviewer on this comment. We have therefore expanded on this section and added the risk ratio in using IV therapy in pregnancy.

  1. Table 1 - this table outlines the factors that result in up or down-regulation of hepcidin. The authors should also note that pregnancy itself results in a down-regulation of hepcidin, particularly in the 2nd and 3rd trimesters (see van Santen et al. 2013).

Author response: We have added the effect of pregnancy on hepcidin levels.

Minor Comments (formatting related):

  1. Just to note that throughout the manuscript, inappropriate hyphenation of words has occurred. This is likely due to a formatting issue but I wanted to flag this now so that it can re rectified.

Author response: The reviewer is correct in that the inappropriate hyphenation of some words is due to formatting issue. We have now corrected the inappropriately hyphenated words.

  1. The use of sub-headings and section headings appears to be inconsistent throughout the text, with new sections or sub-sections needing to be more clearly distinguished than in their current form (see lines 36, 93 for examples of current issue).

Author response: We have improved the sub-headings.

Thank you again to the reviewer and the editors for considering our manuscript and for providing this constructive critique. We hope that you find our responses satisfactory.

Reviewer 3 Report

I congretulate you for the manuscript. Your text is very good and you describe in detail all the possible factors involved in anaemia in pregnancy.

I give you some comments which could be used to expand the text:

In addition to measuring iron levels and hemoglobin (Hb) levels, there are several alternative or supplementary markers that can help determine anemia in pregnancy. These markers provide additional information about the underlying causes or severity of anemia. Most of these markers you mention in your text Additionally we can add:

Mean Corpuscular Volume (MCV): MCV measures the average size of red blood cells. It can help differentiate between different types of anemia. In iron deficiency anemia, the MCV is usually reduced.

Red Blood Cell Distribution Width (RDW): RDW is a measure of the variation in size of red blood cells. It can help differentiate between different types of anemia. An elevated RDW suggests a mix of both large and small red blood cells, which can be seen in certain types of anemia.

Serum Transferrin Receptor (sTfR): Transferrin receptor levels increase when the body needs more iron. Measuring sTfR levels can help differentiate between iron deficiency anemia and anemia of chronic disease.

Total Iron-Binding Capacity (TIBC): TIBC measures the total amount of iron that can be bound by transferrin. It indirectly reflects transferrin levels. Elevated TIBC can be seen in iron deficiency anemia.

These additional markers can provide valuable information when evaluating anemia in pregnancy and help determine the underlying cause and severity.

There are automated methods available for measuring hepcidin levels. Traditional methods for measuring hepcidin involved labor-intensive and time-consuming techniques as you wrote -  enzyme-linked immunosorbent assay (ELISA) or mass spectrometry.

However, more recently, automated methods have been developed to simplify and streamline hepcidin measurement. One commonly used automated method is the competitive enzyme-linked immunosorbent assay (cELISA), which utilizes specific antibodies to detect and quantify hepcidin in a patient's blood sample. This method allows for relatively quick and efficient measurement of hepcidin levels.

Author Response

Reviewer 3

Comments and Suggestions for Authors

I congretulate you for the manuscript. Your text is very good and you describe in detail all the possible factors involved in anaemia in pregnancy.

I give you some comments which could be used to expand the text:

Thank you to the reviewer for the compliment and the kind word.

In addition to measuring iron levels and hemoglobin (Hb) levels, there are several alternative or supplementary markers that can help determine anemia in pregnancy. These markers provide additional information about the underlying causes or severity of anemia. Most of these markers you mention in your text Additionally we can add:

Mean Corpuscular Volume (MCV): MCV measures the average size of red blood cells. It can help differentiate between different types of anemia. In iron deficiency anemia, the MCV is usually reduced.

Author response: Thank you to the reviewer for the comment and we agree that it is important to add the MCV test  as it is a test also rely on using ratio to differentiate between different types of anaemia. We have added the MCV

Red Blood Cell Distribution Width (RDW): RDW is a measure of the variation in size of red blood cells. It can help differentiate between different types of anemia. An elevated RDW suggests a mix of both large and small red blood cells, which can be seen in certain types of anemia.

Author response: Thank you again to the reviewer for their comment and we agree that RDW test is useful. However, assessment of the normal values are instrument and population dependent and therefore the normal values vary between test laboratories. Therefore, we felt that this test will not be appropriate to add to this manuscript. Hope that this is acceptable by the reviewer.

Serum Transferrin Receptor (sTfR): Transferrin receptor levels increase when the body needs more iron. Measuring sTfR levels can help differentiate between iron deficiency anemia and anemia of chronic disease.

Aauthor response: We agree with the reviewer and we have included the sTfR test in the conclusion section (Line 400) as this is a novel approach which may be implemented in future diagnostic testing.

Total Iron-Binding Capacity (TIBC): TIBC measures the total amount of iron that can be bound by transferrin. It indirectly reflects transferrin levels. Elevated TIBC can be seen in iron deficiency anemia.

Author response: We agree with the reviewer and added the TIBC measurement (Line 205)

These additional markers can provide valuable information when evaluating anemia in pregnancy and help determine the underlying cause and severity.

There are automated methods available for measuring hepcidin levels. Traditional methods for measuring hepcidin involved labor-intensive and time-consuming techniques as you wrote -  enzyme-linked immunosorbent assay (ELISA) or mass spectrometry.

However, more recently, automated methods have been developed to simplify and streamline hepcidin measurement. One commonly used automated method is the competitive enzyme-linked immunosorbent assay (cELISA), which utilizes specific antibodies to detect and quantify hepcidin in a patient's blood sample. This method allows for relatively quick and efficient measurement of hepcidin levels.

Author response: We agree with the reviewer and added the c-ELISA test to the diagnostic section (Line 176).

Thank you again to the reviewer and the editors for considering our manuscript and for providing this constructive critique. We hope that you find our responses satisfactory.

Round 2

Reviewer 2 Report

Authors have addressed my comments, thank you.